# Antifungal Potential of *Bacillus velezensis* CE 100 for the Control of Different *Colletotrichum* Species through Isolation of Active Dipeptide, Cyclo-(D-phenylalanyl-D-prolyl)

**DOI:** 10.3390/ijms23147786

**Published:** 2022-07-14

**Authors:** Tae Yoon Kim, Seo Hyun Hwang, Jun Su Noh, Jeong-Yong Cho, Chaw Ei Htwe Maung

**Affiliations:** 1Department of Plant Sciences, University of California, Davis, CA 95616, USA; treee757@gmail.com; 2Department of Agricultural and Biological Chemistry, Environmentally-Friendly Agricultural Research Center, College of Agriculture and Life Sciences, Chonnam National University, Gwangju 61186, Korea; hsh9477@naver.com (S.H.H.); noo1898@naver.com (J.S.N.); 3Department of Food Science and Technology, College of Agriculture and Life Sciences, Chonnam National University, Gwangju 61186, Korea

**Keywords:** *Colletotrichum* species, *Bacillus velezensis*, dipeptides, conidial germination inhibition, anthracnose

## Abstract

*Colletotrichum* species are important fungal pathogens causing anthracnose of tropical and subtropical fruit and vegetable crops. Dual culture assay indicated that *Bacillus velezensis* CE 100 was a strong antagonist against *C. acutatum*, *C. coccodes*, *C. dematium*, and *C. gloeosporioides*. The volatile organic compounds produced by *B. velezensis* CE 100 affected mycelial growth of *Colletotrichum* species tested in our study and caused twisted hyphal structures of all these fungal species. Chloroform crude compounds of *B. velezensis* CE 100 inhibited four *Colletotrichum* species in a concentration-dependent manner and induced severe damage in hyphal morphology of these fungal pathogens, including swelling, bulging, and multiple branching. Moreover, the active cyclic dipeptide, cyclo-(D-phenylalanyl-D-prolyl), was isolated from chloroform crude extract and identified by nuclear magnetic resonance (NMR) and mass spectrometry. The inhibitory effect of cyclo-(D-phenylalanyl-D-prolyl) on conidial germination of *C. gloeosporioides* occurred in a concentration-dependent manner. The conidial germination rate was completely inhibited by a concentration of 3 mg/mL of cyclo-(D-phenylalanyl-D-prolyl). Scanning electron micrographs revealed that the exposure to cyclic dipeptide resulted in seriously deformed hyphae and conidia with shriveled surfaces in dipeptide-treated *C. gloeosporioides*. Therefore, active dipeptide-producing *B. velezensis* CE 100 is a promising biocontrol agent for *Colletotrichum* species causing anthracnose.

## 1. Introduction

Members of the genus *Colletotrichum,* including *Colletotrichum gloeosporioides,* and *C. acutatum,* are mainly responsible for the prevalence of anthracnose in a variety of tropical and subtropical fruit and vegetables crops [1,2,3,4]. Many *Colletotrichum* species are non-host-specific and several species are likely to invade a single host [5]. Based on the perceived scientific and economic importance, this genus was ranked as the eighth most important fungal pathogen [4]. Many *Colletotrichum* species have been documented as seed-borne pathogens and their existence on dead-plant residues is a major source of inoculum for infection [6]. They can be easily disseminated through water splashing and rainfall in conidium form as well as transmitted through the air in ascospore form [7]. Infections initiate from the attachment of a spore on the plant surface, thereby forming an appressorium from germinating conidia and penetrating the cuticle and epidermal layer of the plant [1,8,9]. Like other postharvest fungal pathogens, latent or quiescent infection of *Colletotrichum* species is the main reason for serious postharvest losses [1,10]. Under suitable environmental conditions, the dormancy turns out to be active again to initiate invasion, usually during the ripening, storage, transportation, and marketing stages of the fruit [11,12,13].

General anthracnose symptoms are sunken and necrotic lesions with the formation of tan or salmon-colored conidial masses on the infected area [1,14]. Chemical fungicides such as mancozeb, benomyl, maneb, and chlorothalonil have been commonly used to prevent anthracnose crop damage [15]. Frequent use and high doses of these fungicides can cause the emergence of fungicide-resistant strains and contaminate the environment, particularly the soil and water [16,17,18]. Growing health concerns and recent interest in the performance of antagonistic microorganisms and their active substances have led to a shift from chemical control to environmentally friendly biological control methods [19,20].

Among several groups of antagonistic bacteria, the genus *Bacillus* is well recognized as a representative group for effective biocontrol due to its inhibitory action on plant pathogenic microorganisms by secreting several antimicrobial metabolites as well as high stimulation of plant growth [21,22,23]. These species are widely distributed in nature, including soil, water, plants, and animals [24]. Their tolerance of a wide range of stressful conditions including extreme temperature and pH is impressive. The tolerance is due to the ability to form dormant structures called endospores that can survive for a prolonged period [25]. Due to these unique features, *Bacillus*-based products have been commercialized in the biocontrol market for the control of bacterial and fungal phytopathogens [26].

Antibiosis is a special mode of action of this *Bacillus* group to compete with or defeat different kinds of human and plant pathogenic microorganisms [27]. *Bacillus* species are great producers of structurally diverse antimicrobial compounds through their ribosomal and non-ribosomal synthetic pathways [28,29,30]. Based on their biosynthesis, there are two major groups of these antimicrobial compounds: bacteriocins produced via ribosomal synthesis and lipopeptides produced via non-ribosomal synthesis by large multi-enzymes [31]. Cyclic lipopeptides and other antimicrobial peptides produced by *Bacillus* species have gained much attention due to their inhibitory action on the growth of several pathogenic microorganisms. The lipopeptides including iturins, fengycin, and surfactin act as key players in direct antagonism against plant pathogenic bacteria and fungi [32]. In our previous studies, *B. velezensis* CE 100 was revealed as an effective agent for controlling phytopathogenic fungi, *C. gloeosporioides* and *Botrytis cinerea* through the action of an antifungal tetrapeptide, cyclo-(prolyl-valyl-alanyl-isoleucyl) and methyl hippurate [33,34]. Therefore, the objectives of the present study were to evaluate the efficacy of *B. velezensis* CE 100 in controlling different *Colletotrichum* species and to characterize its active antifungal substance involved in the suppression of *Colletotrichum* species.

## 2. Results

### 2.1. Antagonistic Activity of B. velezensis CE 100 against Phytopathogenic Fungi

The dual culture assay results showed that *B. velezensis* CE 100 inhibited mycelial growth of all tested fungal pathogens. The inhibition rates against *C. acutatum*, *C. coccodes*, *C. dematium*, and *C. gloeosporiodes* were 44.63%, 66.05%, 63.89%, and 56.47%, respectively, at 10 days after incubation (Figure 1 and Figure 2).

### 2.2. Inhibitory Effect of Volatile Organic Compounds (VOCs) Released by B. velezensis CE 100 on the Growth of Phytopathogenic Colletotrichum Species

The VOCs produced by strain CE 100 affected mycelial growth of all the tested fungal pathogens showing an inhibition rate of 38.10%, 33.71%, 25.93%, and 29.59% against *C. acutatum*, *C. coccodes*, *C. dematium*, and *C. gloeosporioides*, respectively (Figure 3). Figure 4 clearly shows that the structures of VOC-affected hyphae of the fungal pathogens were closely twisted as compared to the controls that showed normal and straight structures under light microscopy. 

### 2.3. Effect of Chloroform Crude Extract on Mycelial Growth and Hyphal Morphology of the Colletotrichum Species

As shown in Figure 5, the chloroform crude extracts of strain CE 100 affected the growth of tested fungi. The inhibitory effect was concentration-dependent and increased with increasing concentration of the chloroform crude extract. The highest inhibition rates (43.83% against *C. acutatum*, 68.69% against *C. coccodes*, 59.05% against *C. dematium*, and 49.86% against *C. gloeosporiodes*) were observed when the fungal pathogens were treated with the highest crude extract concentration (1000 mg/L). However, the highest crude extract concentration was not highly effective in controlling the growth of *C. acutatum* when compared to the other three fungal pathogens. Treatment with the lowest concentration (250 mg/L) resulted in the lowest inhibitory rates of all fungal pathogens. Moreover, the mycelia of the treated fungal colonies were densely packed, showing abnormal expansion and lost regular mycelial morphology compared to the control group with normal mycelial development. Microscopic observation revealed serious hyphal alterations of all fungal pathogens including bulging, twisting, irregular extension, and excessive branching due to the exposure of *B. velezensis* CE 100 crude extract compounds (Figure 6). Straight forms of hyphae with the regular branching were observed in the controls of all phytopathogenic fungi.

### 2.4. Identification of the Antifungal Cyclic Dipeptide

The antifungal cyclic dipeptide (30.6 mg) was isolated through the progressive purification of chloroform fraction (4.0 g) of *B. velezensis* CE 100 culture using silica gel- and ODS-MPLC systems. The molecular formula of the dipeptide was determined as C_14_H_16_N_2_O_2_, established by a protonated molecular ion peak at *m*/*z* 245.1289 [M+H]^+^ (calculated for C_14_H_17_N_2_O_2_, *m*/*z* 245.1290, −0.1 mDa). The ^1^H NMR spectrum showed proline moiety corresponding to proton signals of one methine at δ 4.07 (H-2) and four methylene at δ 3.38–3.56 (H-5) and 2.13–1.17 (H-3 and H-4). Additionally, phenylalanine moiety corresponded to proton signals of five methines at δ 7.22–7.31 (H-2′-H-6′), one methylene at δ 3.11–3.21 (H-7′), and one nitrogenated methine at δ 4.46 (H-8′). Proton–proton correlations of phenylalanine and proline moieties were confirmed in the ^1^H-^1^H COSY spectrum (Figure 7, bold lines). The ^1^H NMR result was supported by the ^13^C NMR spectrum, exhibiting 12 carbon signals, including two amide carbonyl carbons at δ 171.0 (C-1) and 167.0 (C-9′). The MS and 1D NMR results suggested that cyclic dipeptide was coupled with phenylalanine and proline moieties. The amino acids were also confirmed by HMBC correlations (Figure 7, arrows). In particular, ^3^J_HC_ correlations (Figure 7, arrows) of δ 4.07 (H-2′)/171.0 (C-1) and δ 4.46 (H-8′)/167.0 (C-9ʹ) were observed in the HMBC spectrum. The MS and 1D NMR results of the antifungal compound isolated in our study were consistent with those of cyclo-(D-phenylalanyl-D-prolyl) reported by Fdhila et al. [35]. ^1^H (500 MHz) and ^13^C (125 MHz) NMR data of cyclo-(D-phenylalanyl-D-prolyl) in CD_3_OD can be found in Table 1.

### 2.5. Inhibitory Effect of Purified Dipeptide on Conidial Germination of C. gloeosporioides and Hyphal Deformation by Scanning Electron Microscopy (SEM)

The purified dipeptide, cyclo-(D-phenylalanyl-D-prolyl), showed a concentration-dependent effect on inhibition of conidial germination of *C. gloeosporioides*. Although the inhibition rate was only 38.82% at the lowest concentration (1 mg/mL) of the dipeptide, the rate increased to 79.25% when the concentration was increased to 2 mg/mL. The conidial germination was completely inhibited (100%) by a concentration of 3 mg/mL of dipeptide 7 h after incubation (Figure 8).

In addition, after 24 h incubation period, the morphology of hyphae and conidia of *C. gloeosporioides* was altered by the exposure to dipeptide as determined by SEM. The degree of hyphal deformation increased depending on the dipeptide concentration. As shown in Figure 9b,c, the exposure of dipeptide affected hyphal appearance, resulting in swollen cells with rugged surfaces or flattened cells. Treatment with the highest concentration of dipeptide resulted in abnormally shaped and wrinkled conidia (Figure 9d). Although very few conidia germinated with 3 mg/mL dipeptide treatment during a long incubation period, extreme swelling of the extended germ tube with a bumpy surface was also observed. Conversely, normal and healthy hyphal cells were observed in the control (Figure 9a).

## 3. Discussion

Due to numerous advantages over chemical biocides including easy degradation and no toxic residues, antibiotic-producing bacteria are recent targets for plant disease control [36]. *Bacillus* species are special candidates owing to their extensive suppression of a wide range of phytopathogens. Recently, several studies revealed that *B. velezensis* is an effective antifungal agent against many pathogenic fungi including *Botrytis cinerea*, *Fusarium verticillioides*, *F. oxysporum*, *F. solani*, *Magnaporthe oryzae,* and *Ralstonia solanacearum* [21,34,36,37,38]. In the present study, the growth of anthracnose causal agents, *C. acutatum*, *C. gloeosporioides*, *C. dematium*, and *C. coccodes,* was restricted by the antagonistic bacterium, *B. velezensis* CE 100. In dual culture tests, strain CE 100 strongly inhibited mycelial growth of these phytopathogens. The release of different antifungal metabolites and fungal-cell-wall-degrading enzymes into an agar medium by strain CE 100 could be the mode of action for the restriction of fungal growth. Bacterial volatiles are distinctive metabolites synthesized during bacterial growth and have been recognized as potent antimicrobial compounds against phytopathogens. These compounds can easily evaporate and diffuse in heterogeneous environments in solids, liquids, and gases [39]. The genus *Bacillus* is a major producer of antifungal VOCs [40]. In different reports, *Bacillus*-originated VOCs have been shown to be effective in regulating the growth of fruit fungal pathogens such as *C. gloeosporioides*, *B. cinerea,* and *Monilinia fructicola*, thereby reducing the fruit’s decay rate [41,42]. Results of our study also pointed out that strain CE 100 produced diverse antifungal VOCs inhibiting the regular growth of four *Colletotrichum* species resulting in reduced growth rate and development of abnormal dense mycelia. Microscopic analysis revealed that the VOC-affected hyphae of the fungal pathogens were profoundly altered.

Moreover, the chloroform crude extract was highly effective in controlling mycelial growth of all *Colletotrichum* species tested in our study resulting in high inhibition rates. The fungal colonies treated with crude extract compounds were closely packed together into condensed mycelial formation and with reduced fungal pigmentation. Microscopic observation confirmed serious structural disintegration and degradation of the fungal pathogens with distorted and bulbous hyphae. These results indicated that the simultaneous action of several antifungal compounds in chloroform crude can restrict consistently the growth of the fungal pathogens and induce alteration of the hyphal structures. Prapagdee et al. [43] also demonstrated that crude antifungal compounds extracted from culture of *B. subtilis* SSE4 were highly effective in controlling the growth of *C. gloeosporioides,* resulting in a reduction in disease severity index of anthracnose on *Dendrobium* leaves. Moreover, our previous study revealed that mycelial growth of *B. cinerea* was strongly inhibited by different concentrations of crude extract from chloroform, ethyl acetate, and n-butanol soluble layer of *B. velezensis* CE 100 culture [34].

The cyclic lipopeptides synthesized by *Bacillus* species are potential biocontrol products owing to their promising performance concerning the suppression of several phytopathogens [44,45]. The lipopeptides such as surfactin, iturin, and fengycin are major bioactive compounds derived from the genus *Bacillus* and they mainly attack cell membranes of the target pathogens through osmotic disturbance due to pore formation and ion leakage, interaction with lipid layers, and structural alteration of the cell membrane, and interference of membrane integrity, disruption, and solubilization with pore formation [46]. Suppression of growth of *Colletotrichum* species by these lipopeptides has been well documented in several reports [47,48,49]. Iturin A purified from *B. amyloliquefaciens* MG3 strongly restricted mycelial growth and spore germination of *C. gloeosporioides*. Moreover, iturin A caused structural alterations in cell membrane of *C. gloeosporioides* and disrupted protein synthesis of the fungal cell, resulting in a decrease in protein content [49]. However, there are limited studies on the antifungal activity of the smallest cyclic dipeptides. In the present study, the active compound was isolated from the chloroform fraction of strain CE 100 using MPLC through silica and ODS columns. According to the spectra of NMR and mass spectrometry, the compound was identified as cyclo-(D-phenylalanyl-D-prolyl). 

Spores are important structures of the fungal life cycle and spore germination is a fundamental process in initiating hyphal growth for invasion of the host plant [50]. Disruption of the spore-germination process by an active compound could be a promising strategy to reduce or halt the fungal invasion. Therefore, the purified dipeptide was evaluated for its inhibitory action on conidial germination of the common anthracnose causal agent, *C. gloeosporioides,* in the present study. The dipeptide inhibited conidial germination in a concentration-dependent manner. At the highest concentration of dipeptide (3 mg/mL), the conidia completely failed to germinate 7 h after incubation. Although the conidia germinated and developed as hyphae at concentrations of 1 mg/mL and 2 mg/mL, hyphal growth was severely affected by the intense action of dipeptide, resulting in deformed hyphal cells with a shriveled surface and abnormal massive growth of twisted and thin hyphae as observed under scanning electron microscopy. Moreover, during a longer incubation period, non-germinated conidia turned into wrinkle-shaped conidia due to the exposure to dipeptide (3 mg/mL). Based on the results of SEM analysis, the exposure of the purified dipeptide seems to seriously affect the hyphal structure of *C. gloeosporioides*, resulting in the collapse of hyphal cells depending on the dipeptide concentration. Our results are in accordance with those of Choub et al. [33] where cyclic tetrapeptide, cyclo-(prolyl-valyl-alanyl-isoleucyl) inhibited spore germination and mycelial growth of *C. gloeosporioides*. Light microscopic analysis revealed abnormal spore shapes and significant hyphal alterations of *C. gloeosporioides* caused by tetrapeptides of *B. velezensis* CE 100. Kuma et al. [51] reported two cyclic dipeptides, cyclo(D-Pro-L-Met), and cyclo(D-Pro-D-Tyr) from *B. cereus* subsp. *thuringiensis* suppressed significantly the growth of three different pathogenic fungi, *F. oxysporum*, *Rhizoctonia solani,* and *Penicillium expansum*. 

## 4. Materials and Methods

### 4.1. Culture Conditions for Bacterial and Fungal Strains

In this study, *Bacillus velezensis* CE 100 (accession number: KACC 81101BP; Korean Agricultural Culture Collection, KACC) was used and this bacterial strain was isolated from the soil of tomato plant [34]. Strain CE 100 was sub-cultured onto tryptone soy agar (TSA) medium and a single colony was used to inoculate again in tryptone soy broth (TSB) medium at 30 °C and 120 rpm for 2 days in a shaking incubator. Then, 2 day-old broth culture of *B. velezensis* CE 100 (1 × 10^7^ CFU/mL) was kept at −80 °C after mixing with a sterile 50% glycerol solution for further experiment. The fungal pathogens *C. acutatum* (KACC 40804), *C. coccodes* KACC 40011, *C. dematium* (KACC 40013), and *C. gloeosporioides* (KACC 40003) were obtained from KACC and sub-cultured on potato dextrose agar (PDA) medium at 25 °C for 7 days.

### 4.2. Antagonistic Activity of B. velezensis CE 100 against Fungal Phytopathogens

#### 4.2.1. Dual-Culture Assay

Antifungal activities of *B. veleznesis* CE 100 against four phytopathogenic fungi (*C. acutatum*, *C. gloeosporioides*, *C. dematium*, and *C. coccodes*) were evaluated by a dual-culture method. A 5 mm-diameter mycelial plug of each 7-day-old fungal colony was placed on one side of a sterile PDA plate and one loopful colony of strain CE 100 was streaked on the other side of the same medium. The PDA medium inoculated with each fungal pathogen alone was used as a control. Three replicates were performed for each fungal pathogen. The plates were incubated at 25 °C for 7–10 days depending on the growth of fungal pathogen. Mycelial growth inhibition was calculated with the following equation, mycelial growth inhibition (%) = R − r/R × 100 (where R is the radius of the fungal colony in the control plate, and r is the radius of the fungal colony in the dual-culture plate) [34].

#### 4.2.2. Volatile Organic Compound (VOC) Inhibition Assay

For the VOC assay, 100 μL of a 2-day-old pre-inoculated broth culture of *B. velezensis* CE 100 (2 × 10^6^ CFU/mL) was spread on a TSA medium. A 5 mm-diameter mycelial plug of from each 7-day-old fungal colony was placed in the center of the PDA medium. Then, the bacterial plate and fungal plate were sealed together facing each other with parafilm. A TSA plate spread with 100 μL sterile distilled water and sealed together with the fungal plate was used for control. Three replicate plates were used for each fungal pathogen and incubated at 25 °C for 5 days. Mycelial growth inhibition was calculated with the following equation, mycelial growth inhibition (%) = D − d/D × 100 (where D is the diameter of the fungal colony in the control plate, and d is the diameter of the fungal colony in the treatment plate). Hyphal alteration of each fungal pathogen caused by VOCs of *B. velezensis* CE 100 was examined under a light microscope.

#### 4.2.3. Antifungal Activities of Chloroform Crude Extract Compounds against Phytopathogens

The chloroform crude compounds were extracted from *B. velezensis* CE 100 culture according to the previous report by Maung et al. [34]. Different amounts of chloroform crude compounds (50 mg, 100 mg, 200 mg) were prepared by dissolving in 200 µL of methanol. Then, the crude compounds were blended with an autoclaved cool PDA medium to obtain different crude concentrations of 250 mg/L, 500 mg/L, and 1000 mg/L. A 5 mm-diameter mycelial plug of each fungal pathogen was inoculated on each PDA medium with a different concentration of crude extract. Each fungal pathogen inoculated on a PDA medium blended with 200 µL of methanol was used as a control. Three replicate plates of each fungal pathogen were incubated at 25 °C for 7 days. Growth inhibition (%) was calculated relative to the control. A small piece of mycelia from each plate was taken to observe hyphal abnormality caused by crude compounds under a light microscope. 

### 4.3. Isolation and Identification of an Antifungal Compound from Chloroform Crude Extract

The crude compounds from chloroform fraction (4.0 g) of *B. velezensis* CE 100 broth culture were separated using medium pressure liquid chromatography (MPLC; Isolera one, Biotage, Sweden) through SNAP KP silica (340 g and 120 g) and octadecylsilane (ODS; SNAP Ultra C18 120 g) columns with a gradient elution of chloroform (CHCl_3_) and methanol (MeOH). The wavelength was set up at 220 nm and 254 nm to detect the compounds. Eight fractions (C1-C8) were obtained from the chloroform fraction through a silica gel column (340 g) with a gradient elution of 0% CHCl_3_ for 5 min → 100% MeOH for 25 min with a flow rate of 150 mL/min. After a preliminary test for antifungal activity against *Colletotrichum* species was conducted, the active fraction C1 (*t_R_* 12.73−13.47 min, 612.4 mg) was selected and fractionated again through a silica gel (120 g) column by a linear gradient of 15% B for 32 min → 25% B for 51.56 min with a flow rate of 25 mL/min to acquire three subfractions (C1a–C1c). Finally, an antifungal compound (*t_R_* 42.40−45.60 min, 30.6 mg, a cream-colored amorphous form) was obtained from a subfraction C1b (*t_R_* 8.32−11.92, 377.4 mg) using a gradient elution of water and acetonitrile (MeCN) (10% MeCN for 12 min → 50% MeCN for 48 min) through ODS C18 column (120 g) with a flow rate of 25 mL/min.

A hybrid ion-trap time-of-flight mass spectrometer (ESI-QToF-MS; Xevo G2-XS QTOF, Waters, Milford, MA, USA) supplied with an electrospray ionization source was used to determine the molecular weight and formula of the antifungal compound. The isolated compound in deuterated methanol (CD_3_OD) was analyzed using an ^unity^INOVA 500 spectrometer (Varian, Walnut Creek, CA, USA; Korean Basic Science Institute in Chonnam National University). The structural determination of the compound was based on ^1^H and ^13^C NMR spectra, which were supported by the ^1^H−^1^H correlation spectroscopy (^1^H−^1^H COSY), heteronuclear multiple-quantum coherence (HMQC), and heteronuclear multiple-bond correlation (HMBC) results. 

### 4.4. Inhibition of C. gloeosporioides Conidial Germination by the Purified Compound

The conidial suspension was prepared by flooding the full sporulated fungal colony of *C. gloeosporioides* with 10 mL of sterile distilled water and after filtering with the sterile miracloth, the number of conidia was counted using a hemocytometer under a light microscope. The final conidia concentration (10^6^ conidia/mL) was prepared by diluting with sterile distilled water and used for conidial germination inhibition assay. Each amount of the purified compound (1 mg, 2 mg, and 3 mg) was prepared in 12 µL of methanol. The compound was then added into an Eppendorf tube containing 100 µL of PDB medium and 100 µL of conidial suspension, and the volume was increased to reach 1 mL with sterile distilled water to make final concentrations of 1 mg/mL, 2 mg/mL, and 3 mg/mL. The tube containing 12 µL methanol instead of the compound was used as the control. Three replications were used for each concentration of the compound. The tubes were incubated at 25 °C for 7 h to check conidial germination and germination inhibition (%) was calculated based on the control. The tubes were then kept at 25 °C for 24 h to observe the intensity of hyphal alteration affected by different concentrations of the compound using scanning electron microscopy.

### 4.5. Preparation of Samples for Analysis of Scanning Electron Microscopy (SEM)

After 24 h incubation at 25 °C, the conidia and hyphal cells were treated with a 4% glutaraldehyde fixative solution prepared in phosphate buffer for 30 min. After centrifugation, the pellets were resuspended in 1% osmium tetroxide for 1 h. The samples were rinsed two times with phosphate buffer and dehydrated with a series of ethanol solutions (35%, 50%, 75%, 95%, and 100%). Finally, hexamethyldisilazane (HMDS) was used to dry the conidia and hyphal cells completely and the samples were coated with gold particles at 60 °C for 15 min. The morphological changes of fungal pathogen were observed using field-emission scanning electron microscopy (FESEM; Gemini 500, Zeiss, Germany) and energy disperse spectroscopy (EDS, Oxford X-MaxN 80, Cambridge, UK).

### 4.6. Statistical Analysis

The experimental data in this study were compared by the least significant difference (LSD) test at *p* < 0.05 level using analysis of variance (ANOVA) in SAS software version (9.4) (SAS Institute, Cary, NC, USA).

## 5. Conclusions

*Bacillus velezensis* CE 100 exhibited antifungal activity against four *Colletotrichum* species through the release of VOCs and other active substances. Due to the synergistic effect of the crude compounds in chloroform fraction, mycelial growth of *Colletotrichum* species was severely inhibited, leading to serious structural deformations including swelling, bulging, twisting, and abnormal branching in hyphae treated with crude extract as observed under light microscopy. The purified cyclic dipeptide, cyclo-(D-phenylalanyl-D-prolyl) effectively inhibited conidial germination of *C. gloeosporioides* in a concentration-dependent manner. SEM analysis revealed that prolonged exposure to cyclic dipeptide induced severe damage to hyphal cells with shriveled membrane. In further studies, it would be interesting to investigate the inhibitory action of cyclo-(D-phenylalanyl-D-prolyl) on the growth of other phytopathogenic fungi and to identify more effective antifungal compounds from the culture of *B. velezensis* CE 100. *Bacillus*-derived antifungal dipeptide, cyclo-(D-phenylalanyl-D-prolyl), could be used as a template for the development of safe and effective bio-fungicide products and *B. velezensis* CE 100 producing bioactive compounds is a potential biocontrol agent for anthracnose in ecofriendly agriculture. 

## Figures and Tables

**Figure 1 ijms-23-07786-f001:**
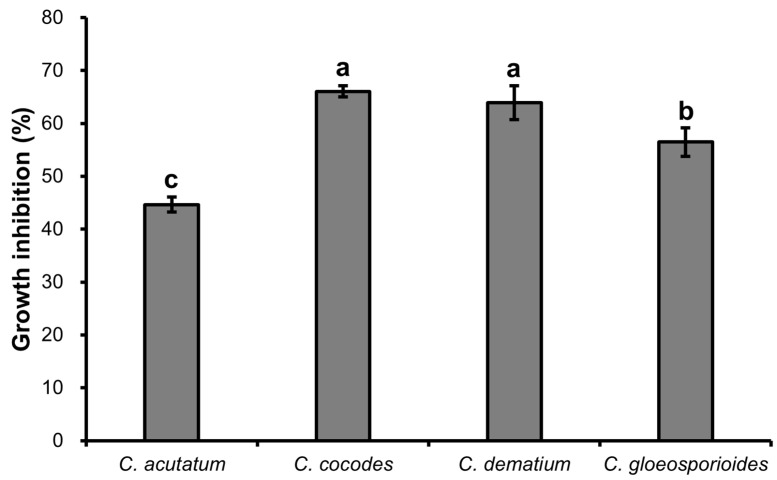
Inhibition of the growth of various phytopathogenic *Colletotrichum* species by *B. veleznesis* CE 100. Histograms represent the mean ± standard deviation of three replications. Different letters represent significant differences (LSD test, *p* < 0.05).

**Figure 2 ijms-23-07786-f002:**
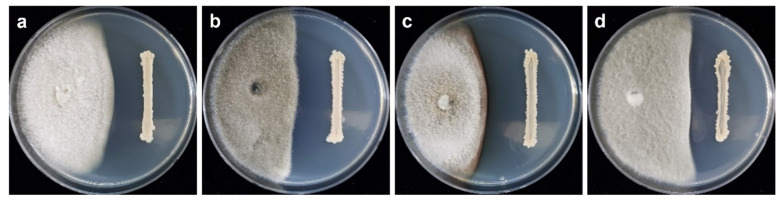
Antagonistic activity of *B. velezensis* CE 100 against *C. acutatum* (**a**), *C. coccodes* (**b**), *C. dematium* (**c**), and *C. gloeosporioides* (**d**).

**Figure 3 ijms-23-07786-f003:**
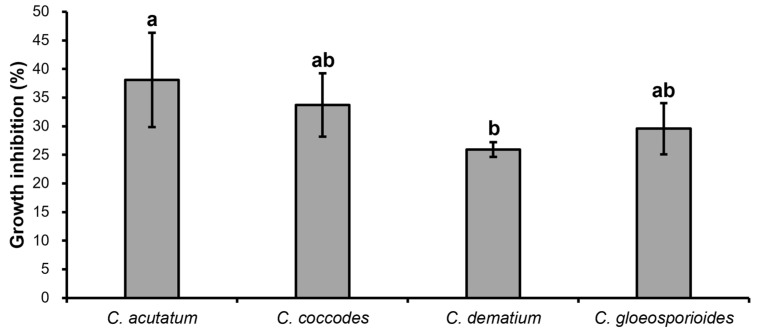
Inhibition of mycelial growth of various *Colletotrichum* species by volatile organic compounds of *B. velezensis* CE 100. Histograms represent the mean ± standard deviation of three replications. Different letters represent significant differences (LSD test, *p* < 0.05).

**Figure 4 ijms-23-07786-f004:**
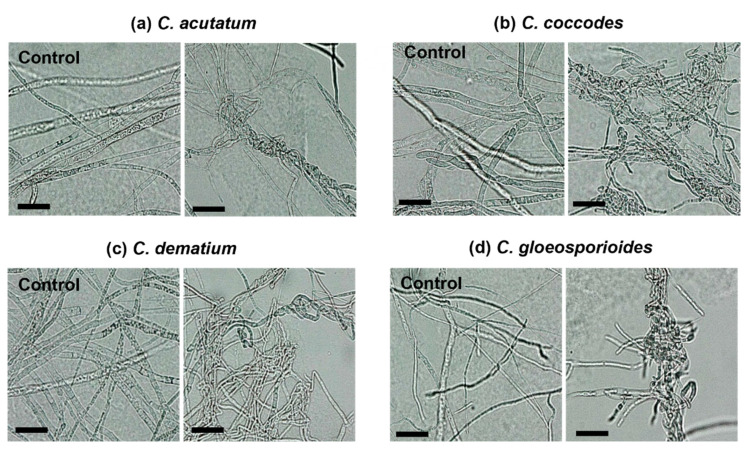
Morphology of VOC-affected hyphae of *C. acutatum* (**a**), *C. coccodes* (**b**), *C. dematium* (**c**), and *C. gloeosporioides* (**d**) compared with morphology of the controls. (scale bar = 25 μm).

**Figure 5 ijms-23-07786-f005:**
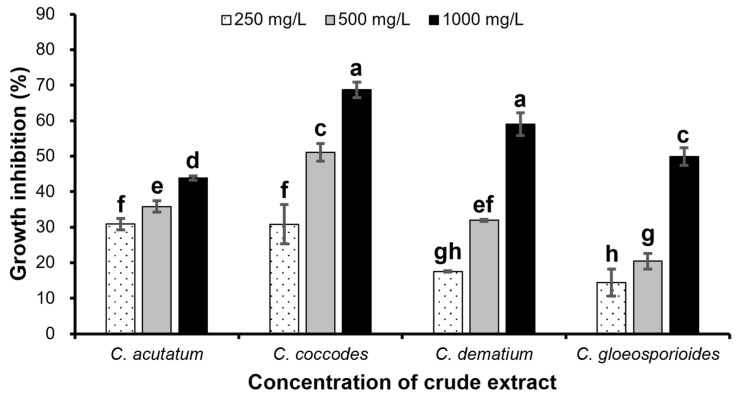
Inhibition of mycelial growth of different *Colletotrichum* species by chloroform crude extract of *B. velezensis* CE 100. Histograms represent the mean ± standard deviation of three replications. Different letters represent significant differences (LSD test, *p* < 0.05).

**Figure 6 ijms-23-07786-f006:**
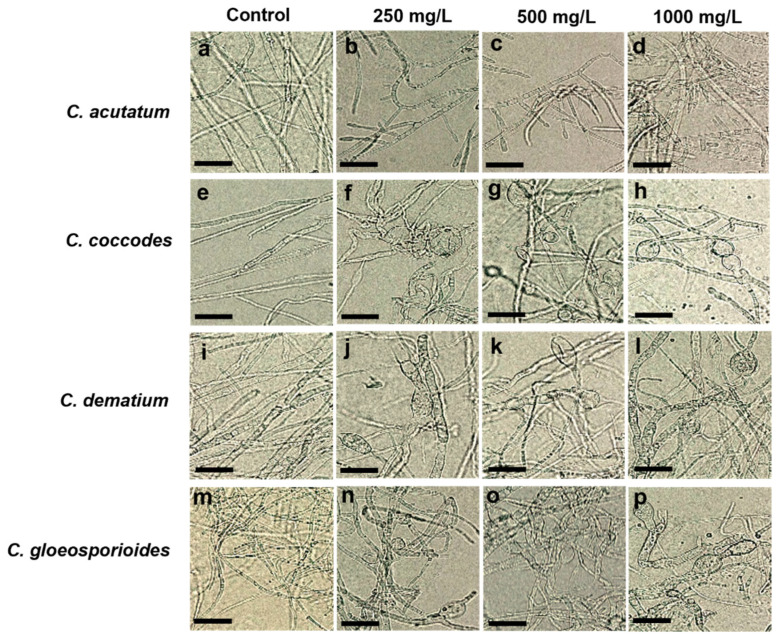
Hyphal deformation of *C. acutatum* (**b**–**d**), *C. coccodes* (**f**–**h**), *C. dematium* (**j**–**l**), and *C. gloeosporioides* (**n**–**p**) caused by 250 mg/L, 500 mg/L, and 1000 mg/L of chloroform crude extract of *B. velenznesis* CE 100 compared to controls (**a**,**e**,**i**,**m**) (scale bar = 25 μm).

**Figure 7 ijms-23-07786-f007:**
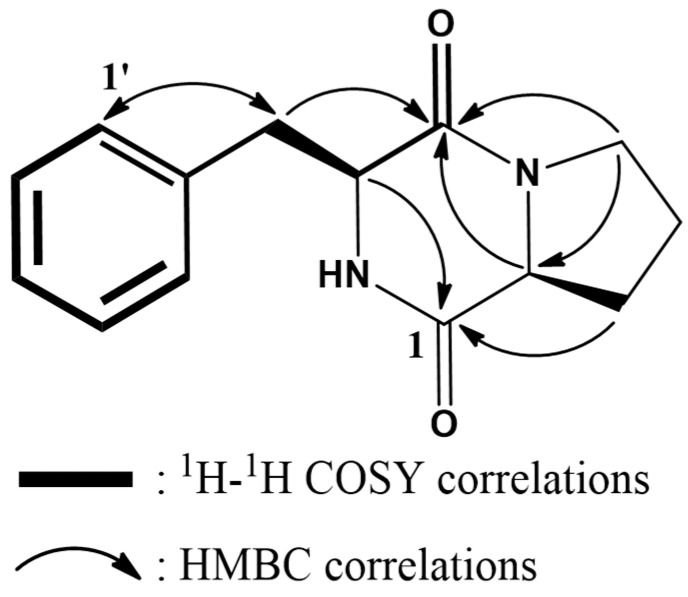
Chemical structure of the purified cyclic dipeptide, cyclo-(D-phenylalanyl-D-prolyl).

**Figure 8 ijms-23-07786-f008:**
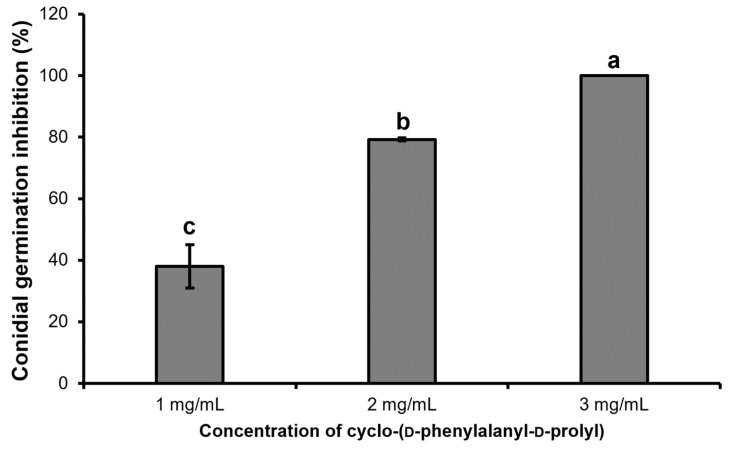
Inhibition of conidial germination of *C. gloeosporioides* by different concentrations of a purified dipeptide, cyclo-(D-phenylalanyl-D-prolyl). Histograms represent the mean ± standard deviation of three replications. Different letters represent significant differences (LSD test, *p* < 0.05).

**Figure 9 ijms-23-07786-f009:**
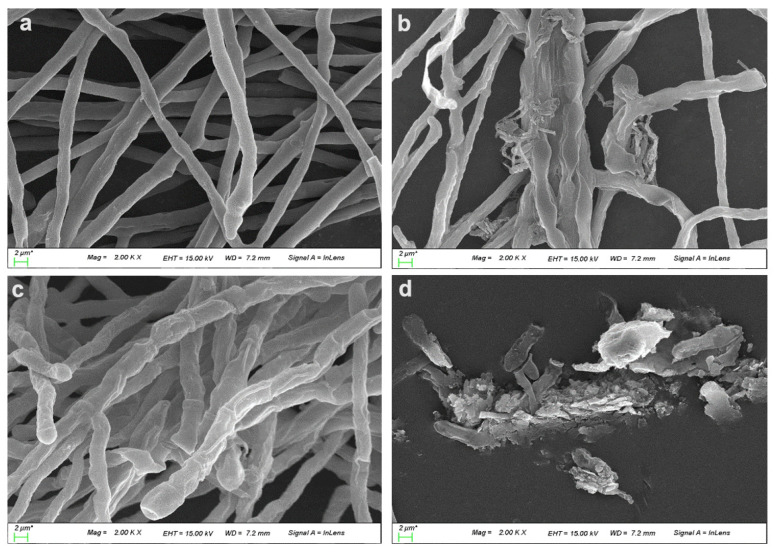
Hyphal deformation of *C. gloeosporioides* caused by 1 mg/mL (**b**), 2 mg/mL (**c**), and deformation of conidia caused by 3 mg/mL of the purified dipeptide, cyclo-(D-phenylalanyl-D-prolyl) (**d**) compared to the untreated control with normal hyphal formation (**a**) at 24 h after incubation as observed under scanning electron microscopy.

**Table 1 ijms-23-07786-t001:** ^1^H (500 MHz) and ^13^C (125 MHz) NMR data of cyclo-(D-phenylalanyl-D-prolyl) in CD_3_OD.

Position	δ_H_ (*Int.*, *Multi.*, *J* in Hz)	δ_C_
1	-	171.0
2	4.07 (1H, ddd, 11.5, 6.0 2.0)	60.2
3a 3b	2.07–2.13 (1H, m) 1.17–1.26 (1H, m)	22.9
4	1.78–1.84 (2H, m)	29.5
5a 5b	3.52–3.38 (1H, m) 3.56–3.40 (1H, m)	46.1
1′	-	137.4
2′,6′	7.22–7.31 (5H, m)	131.2
3′,5′		129.6
4′		128.2
7′	3.11–3.21 (2H, m)	38.4
8′	4.46 (1H, t, 5.0)	57.8
9′	-	167.0

## Data Availability

No applicable.

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
