# Peer review of "Antifungal Potential of Bacillus velezensis CE 100 for the Control of Different Colletotrichum Species through Isolation of Active Dipeptide, Cyclo-(D-phenylalanyl-D-prolyl)"

_ijms, 2022, doi:10.3390/ijms23147786_

Round 1

Reviewer 1 Report

Discussion refer to antifungal compounds (crude extract and lipopeptides) should be elaborated. Other comments are included in attached file.

Reviewer 2 Report

In this study the Authors tested the inhibitory activity of a crude extract of Bacillus velezensis on the mycelial growth of several Colletotrichum species . Moreover they purified a VOC compound from this crude extract and determined its nature as a dipeptide. They tested the inhibitory activity of this dipeptide on conidium germination of Colletotrichum gloeosporioides. The toxic effect of the crude extract and of the dipeptide was also determined by observing the morphological alterations induced on hyphae under the microscope.

The subject per se is of scientific interest but the article has many criticisms:

- The title has to be changed (see suggestions in the text, attached file)

- The English style and the language (technical terms) has to be substantially improved

- The cited literature has to be expanded and updated (see Suggestions in the text): in particular, in the Introduction the Authors have to make clear the criterion they used to choose the species of Colletotrichum they tested

- A major concern is  the Discussion of Results: the Authors mention several antifungal volatile compouds but in this study they charcaterized a single compound, may be they refer to other studies performed by other Authors or by themselves but this was not clear to me and it can not be inferred from Results and M&M

- In M&M the Authors used the term crude VOCs but actually they tested a crude extract and did not provide any evidence the extract contained several VOCs

- References do not conform to the instructions for the Authors (the title of journals should be abbreviated)

- For other observations and suggestions see notes in the text

Round 2

Reviewer 2 Report

After the first revision round the article has been substantially improved and the Authors have addressed almost all relevant observations. However additional text editings have to be perforemed before publications (see notes in the text, attached file)
